# Development of Liquid Crystal Cream Containing Germinated Brown Rice

Suekanya Jarupinthusophon [1], Phatlita Preechataninrat [2] and Oraphan Anurukvorakun [2,*]

1   Department of Chemistry, Phranakhon Rajabhat University, Bangkok 10220, Thailand
2   Department of Cosmetic Science, Phranakhon Rajabhat University, Bangkok 10220, Thailand
*   Correspondence: oraphan@pnru.ac.th; Tel.: +66-94-556-8638

**Abstract:** Herbal cosmetics are gaining popularity over chemicals for beauty products. However, natural products are often prone to deterioration. Therefore, stability and compatibility testing are the main examinations of the safety and reliability of herbal cosmetics. Liquid crystal structures provide better application performances than conventional emulsion systems in terms of stability, controlled release, and moisturizing. Rice is the most profitable crop in Thailand, and the skin healing benefits of rice have been reported. Thus, the current study aimed to develop a liquid crystal cream containing germinated brown rice extract (GBR) or biological cosmetic active ingredients, to study its efficacy on hydration effects and moisturizer, to identify the active ingredient content, such as gamma-aminobutyric acid (GABA), phenolic content, and to evaluate the physical properties and stability of the product. Results revealed that GBR contained GABA and a total phenolic content of 273.28 mg GABA/100 g of rice and 2.58 mg GAE/100 g of rice, respectively. The GBR-liquid crystal cream had good stability and moisturizing effect. The developed products were statistically significantly in hydrating more than a counter brand product. The product that increased moisture the most was the GBR-liquid crystal cream. The GBR-liquid crystal cream provided a high-efficiency moisturizing effect and reliability enabling it to be a premium commercial product shortly.

**Keywords:** germinated jasmine rice; liquid crystal; stability

## 1. Introduction

In recent years, herbs and medicinal plants are increasingly in demand for cosmetics due to their gentle action, low toxicity, and high efficiency [1]. The market size of the worldwide herbal beauty product was assessed at USD 83.52 billion in 2021 and is estimated to reach USD 130.2 billion by 2030 [2]. Herbal products have been rooted and incorporated into health services of the Thai culture. It not only has health benefits but it is also acknowledged for its economic and cultural value [3].

Rice is used as a traditional medicinal plant against beriberi, inflammation, gastrointestinal, and skin ailments [4,5]. Rice also provides a source of nutritional benefits, such as thiamine, riboflavin, niacin, vitamin E, iron, zinc, γ-oryzanol, linoleic acid, etc. [6]. Rice germ contains a desirable bio-functional compound, γ-amino-butyric acid or GABA [7,8], which accumulates during the soaking of brown rice in water [9,10]. In addition, phenolics from rice have been reported as highly biologically active compounds as well. The phenolic compounds in germinated brown rice, such as ferulic acid, *p*-coumaric acid, vanillic acid, protocatechuic acid, caffeic acid, chlorogenic acid, and hydroxybenzoic acid, are well known in supplemental forms and as part of anti-aging ingredients [11,12].

However, natural products are frequently subject to degradation, particularly during storage, which results in the loss of active components, and the occurrence of inactive compounds [13]. The stability of natural products should be addressed to assess the formulation's effectiveness [14]. Therefore, the consideration of natural cosmetic stability can support coping with such problems and improving the products [15]. Understanding

the concerns with natural product stability can improve the idea of dealing with stability issues [16]. Each cosmetic that comes on the market and ultimately to the consumer should be safe, durable, and, above all, effective for the entire period of its shelf life and consumer use [17]. Considering cosmetic evaluation is realized, stability and compatibility testing are the main examinations of the safety of the cosmetics. Moreover, the liquid crystal has an emulsion system that is similar to the structure of water and fat found beneath the skin, which allows it to penetrate the skin effectively and reduce irritation that is likely to occur in other emulsion forms [18–20]. Therefore, the formation of the liquid crystal structure can improve the stability, rheological, and moisturizing properties [21,22].

The scientific report on cosmetic products is significant for the formulation of enhanced cosmetics [23]. Thus, the development of the liquid crystal and active ingredients in germinated brown rice would be a proper technique and ingredients for the anti-aging solution, which results in an excellent skin sensory experience and upgraded moisturizing properties of cosmetics. This study is the first to combine natural extracts (germinated brown rice), confirmed with GABA and total phenolic content, using a liquid crystal emulsion (an innovative delivery system) to encapsulate and deliver the active extract compounds to enhance the efficiency of the moisturizing cream.

The ultimate goal of this research was to develop a liquid crystal cream containing germinated brown rice extract and confirm its efficacy by its hydration or moisturizing effects and active ingredient content, such as GABA and phenolic content. Moreover, the stability of the products was evaluated as well.

## 2. Materials and Methods

### 2.1. Germinated Brown Rice Preparation and Extraction

The brown rice in this study was purchased from Ying Charoen Trading, Khaosan Parn Lan Sub-district, Phayakkhaphum Phisai District, Maha Sarakham Province, Thailand, and harvested during May 2021. A slightly modified method from the Chatchavanthatri report [24] was applied for the extraction. Germinated brown rice was produced by soaking brown rice grains in water to promote germination, and GABA accumulates during this process. The ratio of rice to water was 1:3 for 8 h, and the soaking temperature was 30 °C. Additionally, ultrasonic treatment was carried out at 30 °C for 15 min before and after the soaking process, then grounded using a Moulinex to obtain a rice slurry. The slurry was filtered through a filter bag to obtain a rice solution. The rice solution was dried in a tray dryer at 50 °C. The extract was packed in polypropylene plastic bags and stored in a desiccator at room temperature for further use and modification.

### 2.2. Determination of Gamma-Aminobutyric Acid Content

GABA content analysis was carried out using a slightly modified method from the Karladee report [25]. The germinated brown rice extract (3 mg) from Section 2.1 was dissolved with 80% ethanol in a test tube ($18 \times 120$ mm), shaken thoroughly, and then filtered through filter paper (no. 1). The filtered solution was then boiled in a water bath (80 °C) to evaporate the ethanol. The residue was added into 0.5 mL distilled water and centrifuged at 10,000 rpm for 10 min. The floating portion on top was aspirated, and 0.2 mL of 0.2 M borate buffer and 1.0 mL of 6% phenol were added. The standard GABA solution (0.1–0.3 mL) was added to test tubes ($18 \times 120$ mm) with 0.2 mL borate buffer and 1.0 mL of phenol reagent. The solutions were mixed thoroughly and cooled in a cooling bath for 5 min. Next, 0.4 mL of 10–15% NaOCl was added, then shaken vigorously for 1 min and again cooled in a cooling bath for 5 min. Finally, solutions were boiled in a water bath (100 °C) for 10 min and then allowed to cool.

The quantitative analysis of GABA was performed on a microplate reader (Synergy HT, Biotek, Winooski, VT, USA). The measurements were carried out at a wavelength of 630 nm. GABA content was quantified by comparing it with the absorbance values of a standard GABA curve. The standard calibration curve of absorbance against the concentration of GABA was established. Five different concentrations (0.05–0.5 mg/mL) of standard GABA

were prepared. Each concentration was measured in triplicate. The results are reported as mg GABA/100 g rice.

### 2.3. Determination of Total Phenolic Content

The total phenolic content of the extract was determined by a slightly modified Folin–Ciocalteu assay following the published reports from Inchuen et al. [26], Peanparkdee et al. [27], and Chen et al. [28]. The germinated brown rice extract (0.5 mL) was added to the test tubes followed by 9.5 mL of distilled water, 0.5 mL of Folin–Ciocalteu reagent, and then the resulting mixture was incubated at room temperature for 5 min and then neutralized with 2 mL of 10% sodium carbonate solution. The content of the test tubes were mixed thoroughly. After standing for 1 h at room temperature, the absorbance was measured at 730 nm with a Microplate Reader (Biotek, Synergy HT Multi-Detection, Winooski, VT, USA). The results are expressed as mg gallic acid equivalents per gram of dry matter. The quantitative analysis of total phenolic content was performed on a microplate reader (Synergy HT, Biotek, Winooski, VT, USA). The measurements were recorded at a wavelength of 730 nm. The total phenolic content of the extract was quantified by comparing it with the absorbance values of a standard gallic acid curve. The standard calibration curve of absorbance against the concentration of gallic acid was established. Five different concentrations (100–2000 mg/mL) of standard gallic acid were prepared. Each concentration was measured in triplicate. The results were reported as gallic acid equivalent (mg GAE/100 g rice).

### 2.4. Liquid Crystal Cream Formulation

The formulation strategy for this research was to use Montanov202 (a natural, non-ionic, oil-in-water emulsifier that is palm oil-free) to create liquid crystals. Liquid crystal creams were formulated by mixing phase A (aqueous phase), phase B (emulsifier), phase C (Sepiplus 400), and phase D (skin conditioning agent, preservative, and fragrance) separately. Afterwards, phase A was gently added to phase B under the heated condition at 70–75 °C and using a homogenizer at 6000 rounds/minute for 8 min. Then, phase C was added into the homogenizer and continually mixed for 2 min. After cooling down to 40 °C, the homogenizer's speed was adjusted to 800 rounds/minute for 30 min. Lastly, phase D was added and continually mixed with the homogenizer using the same homogenizing speed for 10 min and the final pH of the product was adjusted to 5.00. Liquid crystal creams were prepared as two formulations, without and with germinated brown rice extract (1.00–1.20% *w/w*) as F1 and F2, respectively. The proper liquid crystal cream for this work was a cream, which absorbed quickly and did not leave a greasy feeling on the skin, and felt light-weight, supple, and soft to the touch. Two liquid crystal creams were prepared using the ingredients and percentages listed in Table 1.

**Table 1.** The ingredients, percentages and function listed.

| Phase | Ingredient | Formula % *w/w* | | Function |
|:---:|:---|:---:|:---:|:---:|
| | | **F1** | **F2** | |
| A | DI-Water | 65–75 | 65–75 | Solvent |
| A | Disodium EDTA | 0.05–0.10 | 0.05–0.10 | Chelating agent |
| A | Glycerin & propylene glycol | 4–6 | 4–6 | Humectant |
| A | Dub diol | 2–4 | 2–4 | Preservative |
| A | Germinated jasmine rice extract | - | 1.00–1.20 | Active Ingredient |
| B | Montanov 202 | 5–6 | 5–6 | Emulsifier |
| B | Emogreen L19 | 5–8 | 5–8 | Emollient, re-fatting agent |
| B | Behenyl alcohol 65 | 5–7 | 5–7 | Emulsifier |
| C | Sepiplus 400 | 0.30–0.60 | 0.30–0.60 | Emulsifier |
| D | Microcare (PHC) | 0.50–0.60 | 0.50–0.60 | Preservative |
| D | Aquaxyl | 0.80–1.20 | 0.80–1.20 | Strengthens the skin barrier, reduces cracking |
| D | Actosome Inoceramide E06 | 0.40–0.60 | 0.40–0.60 | Skin conditioning agent |

### 2.5. Liquid Crystal Inspection

The liquid crystal structure is a process of mixing oil and water phases till homogeneous. In other words, the state is halfway between solid and liquid, with liquid crystal emulsions having a "lamellar structural" arrangement similar to human skin. The developed liquid crystal products without and with germinated brown rice extract (F1 and F2) and a liquid crystal cream from a counter brand cream (F3) were submitted for inspection by polarized light microscopy (PLM) (PLM, Olympus, BX53, Tokyo, Japan) at the Chulalongkorn University Scientific and Technological Research Equipment Centre to detect the "Maltese cross" which is a characteristic configuration of liquid crystals.

### 2.6. Accelerated Stability Studies

The stability study was performed using a slightly modified method based on the publication of Estanqueiro et al. [29]. An accelerated stability study was conducted under the heating/cooling process. The heating/cooling process was achieved by storing the developed products in a fridge/incubator. The storage temperatures changed between $4 \pm 2 \,°C$ and $45 \pm 2 \,°C$ every 24 h for each cycle. The stability study was conducted for six cycles. The stability in ambient temperatures was examined for six cycles as well. Then, 10 g of the developed products from each stored cycle, were submitted to a centrifuge at 3000 rpm for 30 min (Centrifuge, Benchtop Rotofix 32A, Bangkok, Thailand). The phase separation and the changes were reported. Additionally, viscosity and pH values were investigated and reported using a viscometer (DV2T Viscometer Brookfield, Metek, Bangkok, Thailand) and a pH meter (Bench Top pH Meter, ST300 Ohaus, Baesweiler, Germany) to demonstrate the stability of the developed products.

### 2.7. Determination of the Moisture

Liquid crystal creams with and without germinated brown rice extract as F1 and F2 and the counter brand (liquid crystal cream) F3 were evaluated for their efficacy by their hydration and moisturizing effect. The hydration effect of the developed products was measured with a Coreneometer (Corneometer$^®$ CM 825, Cologne, Germany). The measurement sites were $2 \times 2 \, cm^2$, drawn on the pigskin. Three repeated measurements were performed at each test (before and after applying the products) and means and standard deviations were reported. Pigskin is a well-accepted skin model as its integument. It is morphologically and functionally similar to human skin. Paired samples *t*-test (SPSS; Version 23.0) was used to indicate a statistically significant difference in the moisture content between the developed product and a counter brand cream.

## 3. Results

### 3.1. Germinated Brown Rice Preparation and Extraction

The percentage yield of the germinated brown rice extraction was calculated by the following Equation (1): The percentage yield of the extraction was 22.45%.

$$\text{Percentage of yield} = \frac{\text{crude extract of the brown rice (g)}}{\text{initial weight of the broken brown rice (g)}} \times 100 \qquad (1)$$

The possibility of using the extract as an ingredient in the cosmetic industry was revealed with the very high percentage of the extract. The appearance of the extracted powder is shown in Figure 1. The appearance of the extracted germinated brown rice was a light-yellow powder.

### 3.2. Determination of Gamma-Aminobutyric Acid Content

Quantitative analysis of the GABA of the germinated brown rice extract was quantified by comparing the absorbance values to a standard GABA curve ($y = 2.355x - 0.1605$, $r^2 = 0.9965$). A linear fit of the data demonstrated good linearity and reliability of the

quantitative analysis. The results found that the germinated brown rice extract had a GABA content of 273.28 mg GABA/100 g of rice.

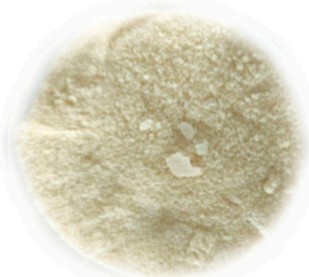

**Figure 1.** The appearance of the extracted germinated brown rice.

### 3.3. Determination of Total Phenolic Content

Quantitative analysis of total phenolic content of germinated brown rice by the Folin–Ciocalteu method was quantified by comparing the absorbance values to a standard gallic acid content curve (y = 75.508 − 0.2176, $r^2$ = 0.9874). A linear fit of the data revealed good linearity and reliability of the quantitative analysis. The results found that the germinated brown rice extract had a total phenolic content of 2.58 mg GAE/100 g rice.

### 3.4. Liquid Crystal Cream Formulation

The acceptance criteria, which were considered successful for the developed liquid crystal cream, were absorbed quickly, did not leave a greasy feeling on the skin, and felt light-weight, supple, and soft to the touch. Liquid crystal creams were successfully developed and are presented in Figure 2 as two formulations to be without and with germinated brown rice extract as F1 and F2, respectively.

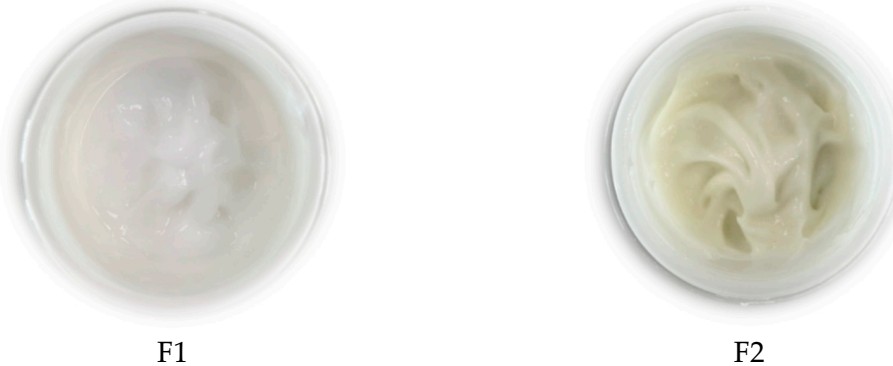

F1                                             F2

**Figure 2.** The appearance of liquid crystal creams: without (F1) and with germinated brown rice extract (F2).

### 3.5. Liquid Crystal Inspection

The results from polarized light microscopy (PLM) (PLM, Oympus, BX53, Tokyo, Japan) are represented in Figures 3 and 4, revealing that the developed products, F1 and F2, showed Maltese cross characteristics and good dispersion of the liquid crystal structure. Interestingly, the liquid crystal cream from the counter brand (F3) did not create a distinct morphology of the Maltese cross. The presence of a Maltese cross indicates the formation of liquid crystals, a molecule arranged in a film encapsulated by the internal droplet cycle, acting as a bumper to prevent direct contact with the internal phase and stabilizes the developed products. The results implied that the developed cream should be able to encapsulate the active substance and be more stable than the comparative counter brand cream.

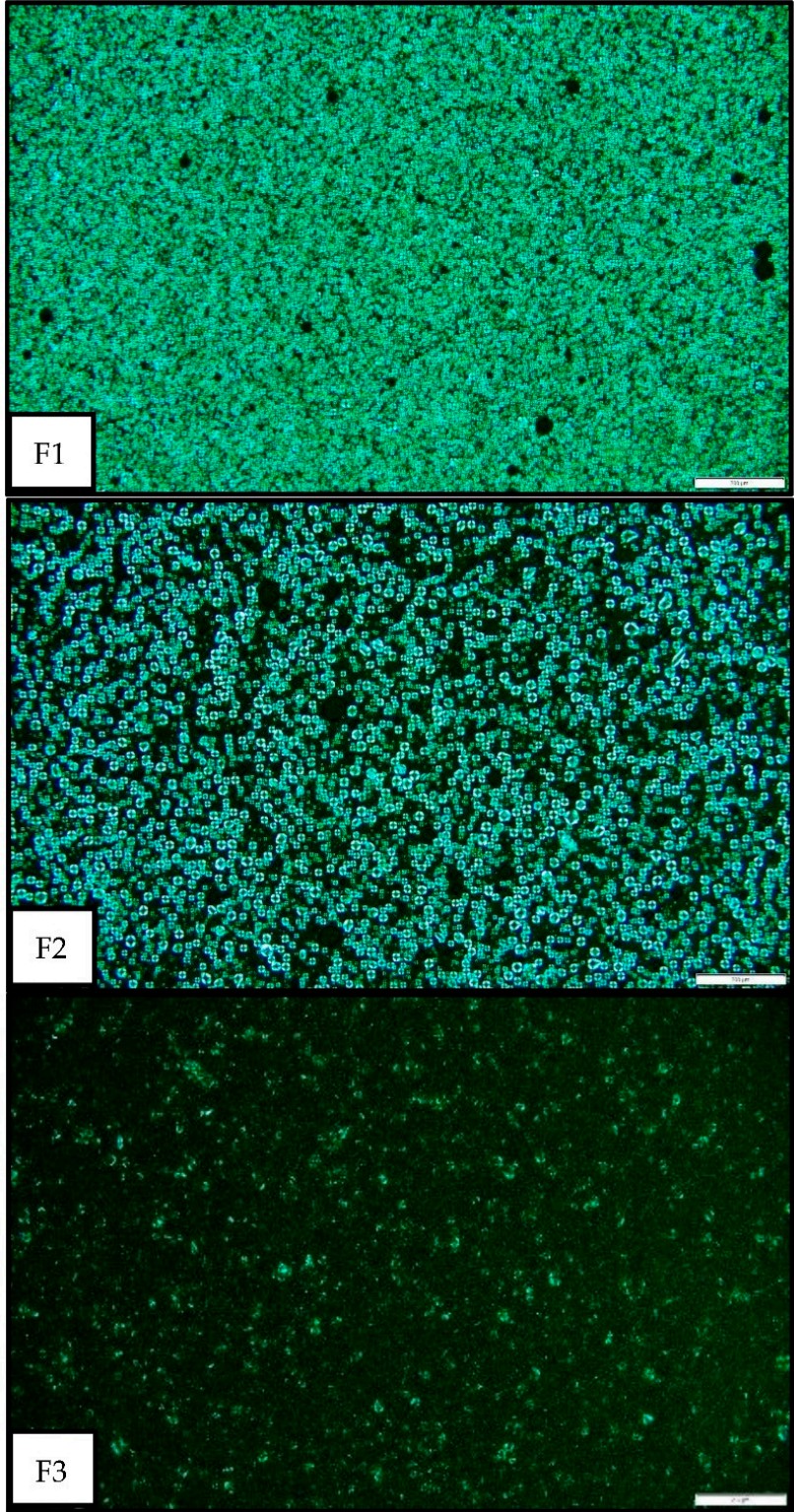

**Figure 3.** Polarized light microscopy photomicrographs (100× magnification) of the developed products (without and with germinated brown rice extract F1 and F2, respectively) and a liquid crystal cream from a counter brand (F3).

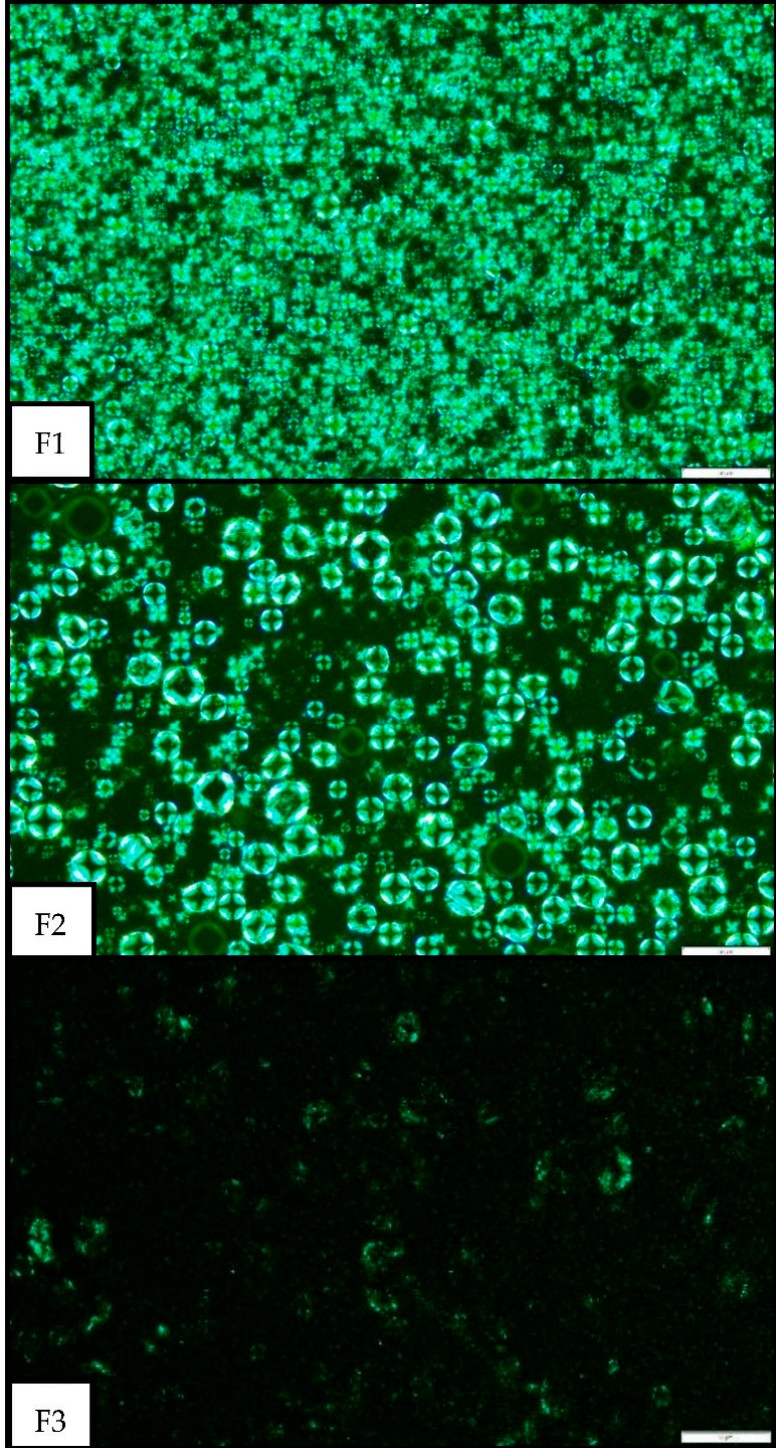

**Figure 4.** Polarized light microscopy photomicrographs (400× magnification) of the developed products (without and with germinated brown rice extract F1 and F2, respectively) and a liquid crystal cream from a counter brand (F3).

*3.6. Accelerated Stability Studies*

Samples subjected to changes in different temperatures can reveal instability quicker than samples stored continuously in one condition. Therefore, the accelerated stability test of this study was performed by heating/cooling for six cycles, as mentioned in Section 2.6. The stability of the products was observed in phase separation, viscosity, and pH values. After the stability test, there was no change in the appearance, color, or odor of the developed

creams. The results showed no phase separation for every cycle, confirmed by centrifugation. The pH values of the formulations were within the range of skin pH (5.0–6.0), and the viscosity was suitable for skin application. Moreover, the pH values, and viscosity of the developed products, were slightly changed after the heating/cooling process and are shown in Tables 2 and 3, respectively. The resulting viscosity change was not statistically different ($p$ value > 0.05) from the freshly prepared product containing germinated brown rice extract (F2). However, for the product without germinated brown rice extract (F1), the viscosity was slightly different ($p$ value < 0.05) compared to the freshly prepared product.

**Table 2.** pH values at the different cycles.

| Formula | Temperature (°C) | pH at the Different Cycles | | | | | | | Mean ± SD | Maximum Percentage Changed (%) |
|---|---|---|---|---|---|---|---|---|---|---|
| | | 0 | 1st | 2nd | 3rd | 4th | 5th | 6th | | |
| F1 | 25 | 5.00 | 5.04 | 5.08 | 5.13 | 5.10 | 5.11 | 5.13 | 5.08 ± 0.05 | 2.60 |
| F2 | 25 | 5.00 | 5.06 | 5.06 | 5.10 | 5.12 | 5.12 | 5.09 | 5.08 ± 0.04 | 2.40 |
| F1 | 45-4 | 5.00 | 5.10 | 5.18 | 5.25 | 5.29 | 5.41 | 5.45 | 5.24 ± 0.16 | 9.00 |
| F2 | 45-4 | 5.00 | 5.13 | 5.21 | 5.27 | 5.35 | 5.48 | 5.50 | 5.28 ± 0.18 | 10.00 |

**Table 3.** Viscosity at the different cycles.

| Formula | Temperature (°C) | Viscosity at the Different Cycles | | | | | | | Mean ± SD | Maximum Percentage Changed (%) |
|---|---|---|---|---|---|---|---|---|---|---|
| | | 0 | 1st | 2nd | 3rd | 4th | 5th | 6th | | |
| F1 | 25 | 10,914.05 | 11,523.12 | 10,370.98 | 11,780.28 | 10,743.01 | 10,127.51 | 11,423.06 | 10,983.14 ± 617.77 | 7.96 |
| F2 | 25 | 10,514.23 | 11,229.00 | 10,819.52 | 11,476.03 | 10,366.40 | 10,210.45 | 11,956.34 | 10,938.85 ± 640.42 | 13.71 |
| F1 | 45-4 | 11,452.00 | 10,912.50 | 12,640.81 | 15,439.05 | 17,406.49 | 17,500.45 | 16,047.26 | 14,485.51 ± 2779.14 | 52.81 |
| F2 | 45-4 | 10,023.50 | 10,122.65 | 10,689.03 | 10,130.44 | 10,290.07 | 9081.90 | 8211.05 | 9792.66 ± 849.79 | −18.08 |

### 3.7. Determination of Moisture

The results of moisture analysis on pigskin with a Corneometer probe of the developed products (without and with germinated brown rice extract denoted as F1 and F2, respectively) and a commercial product (F3), are shown in Table 4. The average moisture content significantly increased in hydration at 95% confidence intervals ($p$ value < 0.05) after applying the products, across all products. The developed products and the counter brand cream contained ingredients that increase moisture in the skin. Additionally, Table 5 shows that the mean difference of each product between products' F1:F2 moisture increase was not significantly different at the 95% confidence intervals ($p$ value > 0.05). It was found that the mean difference between product F1:F3 and product F2:F3 moisture increase was statistically significant at 95% confidence ($p$ value < 0.05). In other words, the developed products, F1 and F2, were significantly more hydrating than the over-the-counter product (F3). The product that increased moisture the most was the liquid crystal cream containing germinated brown rice extract (F2). The Liquid crystal cream containing germinated brown rice extract (F2) provided an average increase of 51.36 units of moisturizing.

**Table 4.** Moisture analysis of the developed products and a counter brand liquid crystal cream.

| Average (Corneometer CM 825) | Product F1 | Product F2 | Product F3 |
|---|---|---|---|
| | Arbitrary Units (Mean ± SD) | Arbitrary Units (Mean ± SD) | Arbitrary Units (Mean ± SD) |
| Baseline | 16.70 ± 1.34 | 15.44 ± 0.94 | 19.20 ± 1.13 |
| After applied the product | 65.95 ± 4.84 | 66.80 ± 5.08 | 66.44 ± 3.66 |
| Difference | 49.49 ± 3.79 | 51.36 ± 4.27 | 47.25 ± 2.73 |
| $p$ value | 0.000 | 0.000 | 0.000 |

Note: Number of experiments: 3. Data given as mean ± SD.

**Table 5.** Results of moisture difference comparison between products.

| | Mean ± SD between F1:F2 | Mean ± SD between F1:F3 | Mean ± SD between F2:F3 |
|---|---|---|---|
| (Average ± SD) | 49.59 ± 3.79:51.36 ± 4.27 | 49.59 ± 3.7:47.25 ± 2.73 | 51.36 ± 4.27:47.25 ± 2.73 |
| *p* value | 0.162 | 0.032 | 0.002 |

Note: Number of experiments: 3. Data given as mean ± SD.

## 4. Discussion

Nowadays, several cosmetic products claim that they have appropriate positive effects on the skin, such as anti-aging. However, scientific reports and evidence about the primary compounds that are used in cosmetic preparations are limited and inconsistent. Therefore, scientific evidence would enhance the reliability of natural products used in premium commercial products in the future. This study is the first report to reveal the effects of brown rice extract tested for GABA and phenolic content in a liquid crystal cream.

The amount of GABA was influenced by many factors, including the duration of seed incubating, soaking temperature, soaking time, or in pre-germinated brown rice [30]. The amount of GABA found in brown rice in this research studies was relatively high (273.28 mg GABA/100 g of rice) compared to other research studies, for example, those by Thitinunsomboon et al. 2013, Kittibunchakul et al. 2017, and Komatsuzaki 2007. The amount of GABA produced was 116.88 mg/100 g of rice reported by Thitinunsomboon et al., 2013 [31] using a process of repeated soaking of germinated brown rice (in tap water at 35 °C, for 3 h) and incubation (at 37 °C, for 21 h). The highest GABA content was 29.46 mg/100 g of rice reported by Kittibunchakul et al., 2017 [32] after incubation at 30 °C for 1 h. Moreover, after soaking for 3 h and gaseous treatment for 21 h at 35 °C, the content of GABA in germinated brown rice was 24.9 mg/100 g of rice [33].

The total phenolic content was 2.58 mg GAE/100 g of rice found in this research, which is consistent with previous research reports showing that rice contains phenolic compounds. While Butsat and colleagues found the phenolic content and antioxidant capacity of brown rice were higher than that of milled rice in which the rice with large grains had the lowest phenolic content (42.57 mg GAE/100 g rice) [34]. Tyagi and colleagues reported the total phenolic contents of nine rice varieties with the amounts ranging from 173.59 to 395.85 mg GAE/100 g of rice [35]. Moreover, Wongsa presented that the total phenolic content of crude extract of colored rice samples ranged from 18.2 to 100.0 mg GAE/100 g of rice [36].

The stability results implied that the developed cream containing germinated brown rice extract had better stability than no extract when considering the percentage changes in pH and viscosity. Moreover, the resulting viscosity change was not statistically different (*p* value > 0.05) from the freshly prepared product containing germinated brown rice extract (F2). The stability test results are consistent with the research reported by Hanno et al. [37]. The report presented that rice extracts contained natural surfactants derived from rice with emulsifying properties, which are beneficial for enhancing compatibility and improving stability [38]. Usually, the change in temperature affects the specific material viscosity of the substance. Some textures might change their viscosity by 10% if the temperature is reduced by 1 °C [39].

Impressively, the liquid crystal cream developed with brown rice extract had good stability and a good moisturizing effect compared with the counter brand cream. The successful development could result from encapsulating the extract in a liquid crystal form, thereby increasing the stability and permeability of the products [18–20]. Since the liquid crystal emulsion structure has some components that mimic human skin such as phospholipids and triglyceride fatty acids, which retain important substances for a long time along with the gentle release of essential substances into the skin. Furthermore, brown rice extract could also moisturize and help reduce wrinkles because of various active substances that are beneficial from rice extract [40].

In conclusion, a combination of natural extract (germinated brown rice), confirmed with GABA and total phenolic content, and liquid crystal emulsion for encapsulating and delivering the active extract compounds, presented a high-efficiency moisturizing cream.

Therefore, this work demonstrates liquid crystal creams containing germinated brown rice extract are ready and reliable for use in premium commercial products.

## 5. Patents

Anurukvorakun, O; Jarupinthusophon, S; Preechataninrat, P. Process of formulating liquid crystal cream with a mixture of brown rice extract. Th. Patent PT 2203001610, 29 June 2022.

**Author Contributions:** Conceptualization, O.A. and S.J.; methodology, O.A. and P.P.; software, O.A.; validation, O.A. and P.P.; formal analysis, O.A. and S.J.; investigation, O.A. and P.P.; resources, S.J.; data curation, O.A.; writing—original draft preparation, O.A.; writing—review and editing, O.A. and S.J.; funding acquisition, O.A. and S.J. All authors have read and agreed to the published version of the manuscript.

**Funding:** This research was funded by the Thailand Science Research and Innovation, grant number 169928.

**Institutional Review Board Statement:** Not applicable.

**Informed Consent Statement:** Not applicable.

**Data Availability Statement:** The data presented in this study are available in this article.

**Acknowledgments:** The proposed experiments have been carried out in the Department of Cosmetic Science, Phranakhon Rajabhat University.

**Conflicts of Interest:** The authors declare no conflict of interest.

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
