# Peer review of "Development of Liquid Crystal Cream Containing Germinated Brown Rice"

_applsci, doi:10.3390/app122111113_

Round 1
Reviewer 1 Report
Comments to the Author
The manuscript by Suekanya Jarupinthusophon et al. describes the preparation of an extract form germinated brown rice, the quantification of active ingredients (gamma-aminobutyric acid and phenolic content) in germinated brown rice, the development of the crystal creams without (F1) and with germinated brown rice extract (F2), and the determination of the physical property/stability of 3 products: F1, F2, and the liquid crystal cream from a counter brand cream (F3). Moreover, the efficacy of the 3 liquid crystal creams was evaluated and compared for their efficacy on hydration effect or moisturizer.
In my opinion the manuscript is interesting, however several points should be better clarified and some parts should be reorganized.
In particular:
- Paragraph 2.2: Why the quantitative analysis of GABA was performed in the grounded rice sample? The quantification of GABA content should be directly performed in the germinated brown rice extract (GBR).
- The amount of the germinated brown rice extract used for the preparation of the liquid crystal cream F2 should be indicated in the paragraph 2.4 of the “Materials and Methods” section.
- Paragraph 2.5: The explanation of the structure of the liquid crystal emulsion should be moved to the discussion section.
- Paragraph 3.1: The sentences “The extracted powder was stored in the refrigerator for further analysis. The containers were properly sealed.” should be moved to the “Materials and Methods” section.
- Paragraph 3.2: The author said: “The germinated brown rice extract had a GABA content of 273.28 mg GABA/100 g of rice”. This is the amount of GABA found in brown rice (3 mg of grounded rice sample from the 2.1 section was dissolved with 80% ethanol). Which is the exact amount of GABA in the powder (light yellow powder indicated in Figure 1) of germinated brown rice extract used for the preparation of the liquid crystal cream F2? Different extraction procedures were used. Please indicate the amount of GABA as mg GABA/100 g of rice extract.
- Paragraph 3.2: The comparison of the amount of GABA found in brown rice with values obtained from other researches should be moved to the “Discussion” section.
- Paragraph 3.3: The author indicated that the germinated brown rice extract had a total phenolic content of 2.58 mg GAE/100 g rice. Please specify the exact total phenolic content in the powder (light yellow powder indicated in Figure 1) obtained from the extraction of germinated brown rice used for the preparation of the liquid crystal cream F2 (as mg GAE/100 g rice extract).
- Paragraph 3.3: Literature data on total phenol contents of rice samples should be moved to the “Discussion” section.
- Table 1, that reports the description of the ingredients and percentages of liquid crystal creams, should be move to the “Materials and Methods” section at paragraph 2.4.
- Paragraph 3.4: In my opinion the differences observed in the appearance, color, odor, pH and viscosity of the developed three creams, immediately after preparation, should be described and compared (by the statistical analysis of differences) in this paragraph of the Results section.
- Paragraph 3.6: The comparison with literature data should be moved to the “Discussion” section.
- The discussion section is very poor and needs a considerable revision. Several parts of the Results section should be moved to the Discussion section and the two sections should be completely reorganized.
Author Response
Response to Reviewer 1 Comments
Please write down "Please see the attachment."
We sincerely appreciate all valuable comments and suggestions, which helped us to improve the quality of the article.
Point 1: Paragraph 2.2: Why the quantitative analysis of GABA was performed in the grounded rice sample? The quantification of GABA content should be directly performed in the germinated brown rice extract (GBR).
Response 1: To be honest, the extract by the 2.1 section was dried in a tray dryer and looked like the grounded rice sample as shown in Figure 1. Moreover, we used that part (from the 2.1 section) for the quantitative analysis of GABA. Anyway, I am so sorry that may be misleading you. To be clear, I have changed a bit of this part (to be the germinated brown rice extract (3 mg) from the 2.1 section was dissolved…..)
Point 2: The amount of the germinated brown rice extract used for the preparation of the liquid crystal cream F2 should be indicated in the paragraph 2.4 of the “Materials and Methods” section.
Response 2: I have already indicated the amount of the GBR in the Material and Methods section.
Point 3: Paragraph 2.5: The explanation of the structure of the liquid crystal emulsion should be moved to the discussion section.
Response 3: I have already moved to the discussion section.
Point 4: Paragraph 3.1: The sentences “The extracted powder was stored in the refrigerator for further analysis. The containers were properly sealed.” should be moved to the “Materials and Methods” section
Response 4: I have already deleted that sentences since in the material and methods already have that detail.
Point 5: Paragraph 3.2: The author said: “The germinated brown rice extract had a GABA content of 273.28 mg GABA/100 g of rice”. This is the amount of GABA found in brown rice (3 mg of grounded rice sample from the 2.1 section was dissolved with 80% ethanol). Which is the exact amount of GABA in the powder (light yellow powder indicated in Figure 1) of germinated brown rice extract used for the preparation of the liquid crystal cream F2? Different extraction procedures were used. Please indicate the amount of GABA as mg GABA/100 g of rice extract.
Response 5: I have already indicate the detail in Response 1
Point 6: Paragraph 3.2: The comparison of the amount of GABA found in brown rice with values obtained from other researches should be moved to the “Discussion” section.
Response 6: I have already moved to the discussion section.
Point 7: Paragraph 3.3: The author indicated that the germinated brown rice extract had a total phenolic content of 2.58 mg GAE/100 g rice. Please specify the exact total phenolic content in the powder (light yellow powder indicated in Figure 1) obtained from the extraction of germinated brown rice used for the preparation of the liquid crystal cream F2 (as mg GAE/100 g rice extract).
Response 7: To be clear, I have changed a bit of the extraction part before determination of total phenolic content (to be the germinated brown rice extract (0.5 mL) were added to test tubes followed by 9.5 mL of distilled water….) to obtained from the extraction of germinated brown rice used for the preparation of the liquid crystal cream F2.
Point 8: Paragraph 3.3: Literature data on total phenol contents of rice samples should be moved to the “Discussion” section.
Response 8: I have already moved to the discussion section.
Point 9: Table 1, that reports the description of the ingredients and percentages of liquid crystal creams, should be move to the “Materials and Methods” section at paragraph 2.4.
Response 9: I have already moved to the “Materials and Methods” section”.
Point 10: Paragraph 3.4: In my opinion the differences observed in the appearance, color, odor, pH and viscosity of the developed three creams, immediately after preparation, should be described and compared (by the statistical analysis of differences) in this paragraph of the Results section.
Response 10: I have already added the details by the statistical analysis of differences in Accelerated stability studies section.
Point 11:
Paragraph 3.6: The comparison with literature data should be moved to the “Discussion” section.
Response 11: I have already moved to the discussion section.
Point 12:
The discussion section is very poor and needs a considerable revision. Several parts of the Results section should be moved to the Discussion section and the two sections should be completely reorganized.
Response 12: I have already revised and moved some of the results section to the discussion section per your advice.
Please do not hesitate to contact me if I can provide any additional information concerning this manuscript.
Best regards,
Oraphan

Author Response
Response to Reviewer 2 Comments
Please see the attachment
We sincerely appreciate all valuable comments and suggestions, which helped us to improve the quality of the article.
Point 1: ABSTRACT
The abstract of a good journal paper always end outlining the benefits of the study findings and recommendations as the way forward. The manuscript is missing such points.
Response 1: I have already added the details in the abstract per your advice.
Point 2: INTRODUCTION
Response 2: I have already added the details in the introduction section per your advice.
Point 3: MATERIAL AND METHOD
3.1 Materials and methods are properly written. However, please explain what does it means by ‘cycles’? The storage temperature changed between 4 and 45oC every 24 hours for each cycle is not clearly understood (section 2.6).
Response 3.1: I have already revised the details.
3.2 Please mentioned brand and country of origin for Polarize Light Microscope (PLM) used in section 3.5
Response 3.2: I have already mentioned the details.
3.3 Instead of pig skin, any model of animal that suggested being used which represent similar to human skin?
Response 3.3: Regarding the review of the cosmetic research, I found that the pig skin was the best one to pick up to reveal the moisturizing effect. Other animals such as rabbits, guinea pigs, and mice might be used for testing shampoo, mascara, and other cosmetic products.
Point 4: RESULTS
The comparison with published results must be placed in discussion section and not in the results.
Response 4: I have already moved to the discussion section.
Point 5: DISCUSSION
5.1 The discussion for all parts of the obtained results in this paper requires extensive comparison trends with published results
Response 5.1: I have already moved the comparison to the discussion section.
5.2 The discussion should be explored deeply and discuss what new they bring to science. This completely missing and needs to be completed.
Response 5.2: I have already revised and moved some of the results section to the discussion section per your advice.
Point 6: CONCLUSION
1) Although this section is not mandatory but can be added to the manuscript to summarize the overall research findings. 2) It should include the knowledge gaps, significance of this study, application of the study and recommendations for future work.
Response 6: I have already added the details for conclusion per your advice.
Point 7: REFERENCES
More references to strength the paper is highly recommended.
Response 7: I have already added the references per your advice.
Please do not hesitate to contact me if I can provide any additional information concerning this manuscript.
Best regards,
Oraphan

Round 2
Reviewer 1 Report
All suggested corrections have been made in the revised version of the manuscript